# Single-Cell Omics in Dissecting Immune Microenvironment of Malignant Gliomas—Challenges and Perspectives

**DOI:** 10.3390/cells10092264

**Published:** 2021-08-31

**Authors:** Bozena Kaminska, Natalia Ochocka, Pawel Segit

**Affiliations:** Laboratory of Molecular Neurobiology, Nencki Institute of Experimental Biology, Polish Academy of Sciences, 02-093 Warsaw, Poland; n.ochocka@nencki.edu.pl (N.O.); p.segit@nencki.edu.pl (P.S.)

**Keywords:** malignant gliomas, glioma heterogeneity, glioma associated microglia/macrophages, tumor infiltrating lymphocytes, single-cell RNA sequencing, mass cytometry, immunosuppression, immunotherapy

## Abstract

Single-cell technologies allow precise identification of tumor composition at the single-cell level, providing high-resolution insights into the intratumoral heterogeneity and transcriptional activity of cells in the tumor microenvironment (TME) that previous approaches failed to capture. Malignant gliomas, the most common primary brain tumors in adults, are genetically heterogeneous and their TME consists of various stromal and immune cells playing an important role in tumor progression and responses to therapies. Previous gene expression or immunocytochemical studies of immune cells infiltrating TME of malignant gliomas failed to dissect their functional phenotypes. Single-cell RNA sequencing (scRNA-seq) and cytometry by time-of-flight (CyTOF) are powerful techniques allowing quantification of whole transcriptomes or >30 protein targets in individual cells. Both methods provide unprecedented resolution of TME. We summarize the findings from these studies and the current state of knowledge of a functional diversity of immune infiltrates in malignant gliomas with different genetic alterations. A precise definition of functional phenotypes of myeloid and lymphoid cells might be essential for designing effective immunotherapies. Single-cell omics studies have identified crucial cell subpopulations and signaling pathways that promote tumor progression, influence patient survival or make tumors vulnerable to immunotherapy. We anticipate that the widespread usage of single-cell omics would allow rational design of oncoimmunotherapeutics.

## 1. Introduction

### 1.1. Classification and Molecular Determinants of Gliomas

Gliomas are tumors of the central nervous system (CNS) that originate from neural stem cells, or astrocytic or oligodendrocytic progenitor cells. Diffuse gliomas represent 80% of primary malignant brain tumors. A new integrated classification system was introduced by the World Health Organization (WHO) in 2016, which comprises five glioma types categorized by both tumor morphology and molecular genetic information [1]. The largest group contains diffuse gliomas such as astrocytic tumors of the WHO grade II and III, the grade II and III oligodendrogliomas, and the grade IV glioblastomas. A separate category encompasses the WHO grade III and IV diffuse gliomas of childhood. This new classification defines as a separate group lower grade astrocytomas with a more circumscribed growth pattern, lack of *IDH1/2* (isocitrate dehydrogenase 1 coding genes) alterations, and frequent mutations of *BRAF* (*v-raf murine sarcoma viral oncogene homolog B1* coding for serine/threonine kinase) (i.e., in pilocytic astrocytoma, pleomorphic xanthoastrocytoma) or *TSC1*/*TSC2* (*tuberous sclerosis complex 1/2* encoding hamartin and tuberin) (in subependymal giant cell astrocytoma). Restructuring of the diffuse gliomas classification revealed new subtypes including IDH-wildtype glioblastoma, IDH-mutant glioblastoma, and H3 K27M-mutant diffuse midline glioma [2].

The *IDH1* gene encodes an isocitrate dehydrogenase 1 which catalyzes the oxidative decarboxylation of isocitrate to α-ketoglutarate (α-KG). Mutations in *IDH1/IDH2* result in the substitution of arginine at codon 132 of the IDH1 or codons 140 or 172 of the IDH2 (IDH1-R132, IDH2-R140 or IDH2-R172). IDH1-R132 mutants have dominant-negative, inhibitory effects on a wild-type IDH1, and gain new functions as they reduce α-KG to its (*R*)-enantiomer of 2-hydroxyglutarate (2-HG). Accumulation of 2-HG in cancer cells inhibits Jumonji class histone demethylases and TET (ten-eleven translocation) family of 5-methylcytosine DNA demethylases (reviewed in [3]). These events result in altered histone methylation and the hypermethylation phenotype [4]. More than 70% of WHO grade II and III diffuse gliomas are IDH1-R132 mutants. *IDH1/2* mutations in some gliomas are associated with longer survival and better responses to chemotherapy [5,6]. Glioblastomas (GBM), divided into an IDH-wildtype (about 90% of cases) and IDH-mutant (about 10% of cases) GBMs, account for 70–75% of all diffuse gliomas and have a median overall survival of 14–17 months. IDH-mutant GBMs are usually secondary glioblastomas with a history of prior lower grade diffuse glioma and are more frequent in younger patients. GBMs encompass also the epithelioid glioblastoma type, giant cell glioblastoma, and gliosarcoma, which are more frequent in children and younger adults, and often harbor a *BRAF* V600E mutation.

Comprehensive analyses of genomic transcriptomic alterations in GBMs demonstrated the most common alterations on chromosome 7 (*EGFR/MET/CDK6*), chromosome 12 (*CDK4* and *MDM2*), and chromosome 4 (*PDGFRA*), as well as frequent changes of genes such as *SOX2*, *MYCN*, *CCND1*, and *CCNE2* [7,8]. *EGFR* (epidermal growth factor receptor), *PDGFR* (platelet derived growth factor receptor), and *MET* (MNNG HOS transforming gene) encode receptor tyrosine kinases that control cell proliferation and survival signaling. *NF1* (neurofibromathosis 1), *K-RAS* (KIRSTEN rat sarcoma viral oncogene homologue), and *B-RAF* are components of signaling pathways controlling proliferation and survival. The cell cycle regulatory protein cyclin D1 (*CCND1*), cyclin E2 (*CCNE2*), cyclin-dependent kinases 4 and 6 (*CDK4/6*), MDM2 (*Mouse double minute 2*), and transcription factors SOX2 (*sex-determining region Y-box 2*) and MYCN (N-*MYC* oncogene) control the cell cycle. Several novel mutations or gene rearrangements were found in genes involved in chromatin organization: *SETD2* (*Set domain containing 2*), *ARID2* (*AT-Rich Interaction Domain 2*), *DNMT3A* (*DNA methyltransferase 3 alpha*), *KRAS/NRAS* [9]. Aberrant expression or mutations in genes such as *EGFR, NF1,* and *PDGFRA/IDH1* allowed the delineation of the classical, mesenchymal, and proneural GBM subtypes, respectively. A tumor transcriptional type and specific genetic alterations confer survival advantages. A weak association between the proneural GBM subtype and longer survival has been reported [10]. Correlative analyses of transcriptomic profiles showed the survival advantage of the proneural subtype, which is likely to be due to the Glioma CpG island methylator phenotype (G-CIMP). Methylation of the *MGMT* (O^6^-methylguanine–DNA methyltransferase) gene promoter (resulting in low expression of the DNA repair protein) emerged as a predictive biomarker for treatment response only in a classical GBM subtype [8,10]. The subsequent integration of various data sources, multiple methylation, and gene platforms identified new diffuse glioma subgroups in adults. The IDH-mutant non-codel lower grade gliomas (LGG) and GBM were dissected based on the genome-wide patterns of DNA methylation, into two separate subgroups (G-CIMP-low and G-CIMP-high). This dissection is important as the G-CIMP-low subset had an unfavorable clinical outcome [9].

Introduction of scRNA-seq to GBM studies revealed their considerable genetic heterogeneity and discrete functional states. Implementation of single cell RNA sequencing (scRNA-seq) to glioma studies showed the considerable intratumoral genetic heterogeneity and mosaic expression of genes coding for surface receptors and ligands, including *EGFR*, *PDGFRA*, *PDGFA*, *FGFR1* (*fibroblast growth factor receptor 1*), *FGF1* (*fibroblast growth factor 1*), *NOTCH2* (*notch receptor 2*), and *JAG1* (*jagged canonical notch ligand 1*) [11]. Transcriptomic analyses revealed four meta-signatures related to hypoxia, complement/immune response, oligodendrocytes, and cell cycle. Application of the stemness signature (*POU3F2*, *SOX2*, *SALL2*, *OLIG2*) demonstrated the presence of glioma stem cells. *POU3F2* (*POU Class 3 Homeobox 2*), *SOX2*, *SALL2* (*Spalt Like Transcription Factor 2*), and *OLIG2* (*Oligodendrocyte transcription factor 2*) encode transcription factors related to the maintenance of the stemness phenotype. Mesenchymal tumors with high expression of stemness markers (*POU3F2*, *SOX2*, *SALL2*, *OLIG2*) had significantly worse outcome than the proneural subtype tumors [11]. Integrative studies of scRNA-seq data of GBMs, combined with genomic and bulk gene expression analyses collected in The Cancer Genome Atlas (TCGA), allowed the definition of discrete functional GBM states. Copy number amplifications of the *CDK4*, *EGFR*, and *PDGFRA* loci and mutations in the *NF1* locus define GBM states: neural progenitor-like (NPC-like), oligodendrocyte-progenitor-like (OPC-like), astrocyte-like (AC-like), and mesenchymal-like (MES-like) states. While their relative frequency varies between tumors, genetic alterations in *CDK4*, *PDGFRA*, *EGFR*, and *NF1* define a predominant state [12,13]. *EGFR* aberrations were associated with a predominance of the AC-like cells, while amplifications of *CDK4* and *PDGFRA* were connected with the prevalence the NPC-like and OPC-like states, respectively. Chr5q deletions and *NF1* alterations affected the frequency of MES-like states [12]. Collective data suggest that the molecularly distinct GBM subtypes representing various functional states have a profound impact on their microenvironment and this notion is supported by the data from mouse GBM models showing that well-defined driver mutations may create unique microenvironments [14].

### 1.2. Immunological Uniqueness of the Central Nervous System

The CNS has long been considered an “immune privileged site” due to its unique protein composition and excellent separation from other organs. The presence of the blood–brain barrier (BBB) shields CNS autoantigens from immune recognition and inflammation, and various local tolerance—related mechanisms cooperate in functional silencing or elimination of infiltrating T cells [15,16]. BBB is not an ultimate barrier under pathological conditions and even in physiological conditions rare CD4+ and CD8+ T cells may infiltrate CNS [17]. Moreover, CNS and its meningeal coverings accommodate diverse myeloid cells that encompass parenchymal microglia, CNS border-associated macrophages (BAMs), consisting of perivascular, choroid plexus and meningeal macrophages, and dendritic cells (DC), as well as natural killers (NK) [18,19,20]. The resident myeloid cells constitutively express major histocompatibility complex (MHC) molecules in the steady state and are professional antigen-presenting cells (APC). Microglia express both MHC class I and II molecules, which allows efficient antigen presentation to CD8+ and CD4+ T cells, respectively. A small number of DCs is sufficient to reactivate CD4+ T cells within CNS [18]. Under pathological conditions, immune system cells can migrate to the CNS through the choroid plexus or meningeal vasculature. CD49d/VCAM-1 (vascular cell adhesion molecule 1) facilitates entry of CD8+ T cells into the CNS [17]. After injection of the CD8+ T-cells into the brain, cells with a highly activated CD69^high^ phenotype were gradually lost in the brain parenchyma, whereas CD69^low^ cells that migrated from the spleen persisted longer [17].

Insights from fate mapping studies show that the innate immune cells of the CNS have distinct ontogeny from bone marrow (BM) derived mononuclear phagocytes, and colonize CNS early in the development. In addition, as consistently shown by recent multi-omics and fate-mapping studies, BAMs and microglia are distinct cell populations considering their transcriptomic and proteomic profiles, ability to self-renew, and their development [21,22,23,24,25]. In adulthood, both microglia and BAMs are dependent on constant stimulation of colony-stimulating factor-1 (CSF1) receptors. CSF1R signaling via CSF1 and/or IL34 ligands regulates the production and differentiation of most circulating and tissue-resident macrophages. In both mice and humans an alternative ligand for CSF1R-interleukin (IL)-34 that is produced by neurons is essential for microglia maintenance [26,27,28].

## 2. The Complexity of the Immune Microenvironment of Malignant Gliomas

Tumor microenvironment (TME) is heterogeneous and consists of tumor cells, stromal cells, endothelial cells, innate immune cells (microglia, macrophages, neutrophils, dendritic cells, innate lymphoid cells, myeloid-derived suppressor cells (MDSCs)), NK cells, and adaptive immune cells (T cells and B cells). In addition, there are the non-cellular components of extracellular matrix, in which cells are embedded. TME plays an important role in brain tumor progression, immune tumor evasion, and responses to therapies, all of which determine patient survival [29,30]. Numerous studies have demonstrated that GBMs are infiltrated by immune cells that made up to 30% of a tumor’s mass [31]. The predominant population consists of glioma-associated microglia and macrophages (GAMs) and their numbers inversely correlate with patients’ survival (reviewed in [29,32]). The kinetics studies of accumulation of microglia (CD11b^+^CD45^low^) and macrophages (CD11b^+^CD45^high^) in experimental GL261 gliomas showed that microglia are first attracted to the growing tumor and macrophages are late newcomers to the glioma TME [33].

Such extensive accumulation of innate immune cells in gliomas might be misleading as these events do not reflect the effective anti-tumor immunity. Several mechanisms by which malignant gliomas evade elimination by the immune system include reduced expression of antigen processing and presentation proteins; recruitment of suppressor myeloid and regulatory T cells (Tregs); production of immunosuppressive factors (i.e., prostaglandins), cytokines such as TGF-β1 (transforming growth factor beta 1), and interleukin (IL-10); and up-regulation of ligands for co-inhibitory receptors (i.e., PD-L1, Programmed death-ligand 1) that reduce activities of tumor-infiltrating T lymphocytes (TILs) [34,35]. Reprogramming of glioma infiltrating microglia and macrophages into the tumor supportive, immunosuppressive phenotype (resembling to some extent the in vitro inducible M2 phenotype) has been well documented [31,36,37,38]. Microglia educated by glioma cells promote glioma invasion [39,40,41]. MDSCs produce IL-6 and IL-10, to support their expansion, and numerous cytokines (i.e., TGF-β) and chemokines to attract and induce Tregs. High activity of arginase 1 in MDSCs results in exhaustion of some amino acids necessary to support proliferation of T cells, while released secretory enzymes, their metabolites or membrane transporters, block T cell activation [42] or lead to apoptosis of T cells [43]. Activated DCs contribute to anti-tumor immunity by increased expression of MHC class II, costimulatory molecules and C-C chemokine receptor type 7 (CCR7), and high ability to produce cytokines [44]. However, GBM-derived immunosuppressive factors block DC maturation and reprogram DCs into immunosuppressive or regulatory phenotype [45].

GAMs isolated from GBM specimens had few innate immune functions intact, but their ability to secrete cytokines and upregulate co-stimulatory molecules is not sufficient to initiate anti-tumor immune responses [46]. Despite MHC II expression, GAMs lacked expression of the costimulatory molecules CD86, CD80, and CD40 critical for T-cell activation. This resulted in a lack of effector/activated T cells (glioma infiltrating T-cells were CD8+CD25–). In turn, a prominent population of regulatory CD4+ T cells (CD4+CD25+FOXP3+) infiltrating the tumor was detected [47]. GAMs within the tumor microenvironment release a number of immunosuppressive cytokines and promote tumor growth in a variety of ways: by priming invasion, supporting cancer stem cells, blocking the anti-tumor immunity, and favoring the genetic instability (reviewed in [29,32]).

The issue of cell heterogeneity and functional phenotypes of GAMs (isolated as CD11b^+^ cells from clinical GBM samples or experimental murine gliomas) and analyzed using bulk transcriptomics have been addressed in many studies, but demonstrated conflicting results [38]. Several studies showed the pro-tumor phenotype of GAMs (suppressive M2 phenotype of peripheral macrophages) in rodent gliomas [33,48,49], whereas other studies indicated mixed M1/M2-like phenotypes [50] or M0 phenotype in human glioblastomas [51]. Lineage tracing studies of mouse experimental gliomas clearly demonstrated the significant contribution of BM-derived macrophages to a pool of GAMs [52]. However, in the studies on GAM phenotype the immune populations were isolated as CD11b^+^ cells immunosorted by FACS or magnetic beads combining all myeloid cells present in GBM tissues, including cells from circulating blood. The M1/M2 classification of the macrophage states in TME is now recognized as being oversimplified. Single-cell studies provide compelling evidence that the transcriptional programs expressed by specific subpopulations of GAMs are not uniform and exhibit spatial and temporal distinctiveness. Therefore, a precise dissecting of GAMs’ heterogeneity, reflected by diverse cellular types and states, could not be resolved with classical methods.

## 3. Immune Microenvironment of Malignant Gliomas—Insights from Single-Cell Omics

### 3.1. Operating Principles of Single Cell Technologies

The heterogeneity of the immune infiltrates and specific roles of distinct subpopulations in health and disease could not be solved using traditional methods, which is particularly important in dissecting myeloid subpopulations expressing similar markers. Single-cell studies provided a breakthrough and novel insights into the diversity of immune cells in health and disease.

Single-cell RNA sequencing (scRNA-seq) permits one to determine the entire transcriptome of thousands of individual cells [53]. In recent years, scRNA-seq has been used to study immune cells of the brain both during development and in various CNS pathologies [54,55,56,57,58,59,60,61]. Cytometry by time-of-flight (CyTOF) utilizes monoclonal antibodies conjugated with metal isotopes, which, due to a minimal overlap between channels, allow evaluating more than 40 parameters in a single run. This method significantly outperforms classic flow cytometry analysis or multicolor flow cytometry [62]. Cellular indexing of transcriptomes and epitopes by sequencing (CITE-seq) combines the two approaches by application of oligonucleotide-conjugated antibodies, of which the oligonucleotide tags are sequenced in parallel with transcriptome libraries, allowing simultaneous RNA and surface protein measurements [63]. This relatively new method has an advantage of comparing RNA expression with protein levels that are not always corresponding [63]. It allows to stain a virtually unlimited number of epitopes, although only surface proteins can be targeted. Moreover, spatial transcriptomics techniques are emerging that allow visualization and a quantitative analysis of the transcriptome with spatial resolution in tissue sections. In this approach, histological sections are positioned on arrayed reverse transcription primers with unique positional barcodes and the technique allows visualization of the distribution of mRNAs within tissue sections [64,65].

In the past, more detailed understanding of the characteristics of functional phenotypes of the immune infiltrates in gliomas was hampered by the lack of reliable markers allowing for separation of specific subpopulations. In the present study, we summarized and critically assessed the current knowledge regarding the composition and functions of immune subpopulations that emerges from the single cell omics studies (Table 1).

### 3.2. Functional Phenotypes of the Glioma Associated Microglia and Macrophages

GAMs consist of myeloid cells of two origins: brain-resident microglia and BM-derived macrophages. Microglia originate from hematopoietic precursors—erythromyeloid progenitors (EMP) that develop in early embryonic life in the yolk sack [74]. A CX3CR1+ subpopulation differentiate from EMPs giving rise to microglial progenitors which migrate to the brain starting from the embryonic day 9.5, until formation the blood-brain-barrier [75]. Microglia are long-living cells capable of self-renewal that is independent of the bone marrow and circulating precursors [74,76]. In contrast, infiltrating macrophages originate from circulating monocytes that renew continuously from hematopoietic stem cells (HSC) residing in the bone marrow. HSCs are capable of giving rise to all blood cell lineages such as red blood cells, lymphocytes, and myeloid cells [77]. Despite different ontogeny, cells of the two populations express many common surface proteins and, due to the lack of reliable discrimination markers, their specific roles in gliomagenesis has not been deciphered. Low consistency demonstrated in the GAM transcriptional signature from various bulk RNA-seq studies suggests a substantial heterogeneity of GAMs [38].

#### 3.2.1. GAMs Origin and Localization Influence the Expressed Phenotype

scRNA-seq allows a transcriptome analysis in single cells of distinct cell populations and states. Müller et al. (2017) employed GAM transcriptional profiles that were derived from a genetic lineage-tracing studies [52] to discriminate microglia and macrophages in their scRNA-seq analysis of IDH-wt GBMs [57]. They determined P2RY12 (microglia) and CD49d (macrophages) as good discriminating markers, and demonstrated that although all GAMs show tumor-induced expression of human leukocyte antigen-DR (HLA-DR), its expression is higher in P2RY12- macrophages compared to P2RY12+ microglia. This is in line with a recent study demonstrating significantly increased HLA-DR levels in monocyte-derived macrophages (MDMs) versus microglia both in IDH-wt and IDH-mut gliomas [78]. Müller et al. (2017) have applied the identified transcriptomic signature of microglia and macrophage GAMs to estimate dominant populations across glioma anatomical regions in the dataset from the Ivy Glioblastoma Atlas Project [79]. Analysis of the bulk RNA-seq performed on glioma microdissected regions indicated that microglial GAMs are enriched in the leading edge and adjacent white matter, whereas macrophage GAMs show increased accumulation in the areas containing hyperplastic blood vessels, microvascular proliferation, and peri-necrosis [57]. Consistently, scRNA-seq on human GBM samples resected from a tumor core and tumor periphery demonstrated that macrophages (69%) predominate within the tumor core whereas microglia are most abundant at the tumor edge (86%) [56].

Subsequent transcriptional analysis of periphery- and core-derived samples demonstrated that GAMs in the periphery are enriched in the expression of pro-inflammatory interleukin *IL1B* and a number of cytokines (*CCL2*, *CCL3*, *CCL4*, *TNF*), as well as colony-stimulating factor (*CSF1*) and its receptor (*CSF1R*). Core-derived GAMs present increased expression of pro-angiogenic *VEGFA* (vascular endothelial growth factor A), hypoxia-induced *HIF1A* (hypoxia induced factor alpha), and anti-inflammatory interleukin *IL1RN* [56]. These observations point to the importance of the tumor proximity in the tumor-induced activation, but also suggest differences in transcriptional programs expressed in microglia and macrophages. Interestingly, immune-checkpoint encoding genes *CD274* (PD-L1), *PDCD1LG2* (PD-L2), *CD80,* and *CD86* (CTLA4 receptors) were expressed in both regions, with a slightly higher level in the periphery that is indicative of the immune-suppressed microenvironment [56].

#### 3.2.2. Transcriptional Programs of Glioma-Associated Macrophages

Cell state can be characterized by the expressed transcriptional program, informing about specific cell function. Accumulating evidence from the single-cell studies suggests that various transcriptional programs can be expressed by the same cell type. Simultaneously, expression of one transcriptional program is not necessarily restricted to a single cell type.

Sankowski et al. (2019) performed scRNA-seq analysis of patient-derived GBM samples and normal-appearing brain tissue from epileptic brains [59]. Cell clusters consisting mainly of GBM-derived cells showed decreased expression of microglia core genes, and induced interferon-associated genes (*IFI27*, *IFITM3*), lipid metabolism-related (*LPL*, *APOE, TREM2),* and MHC-I and -II encoding genes (*HLA-A/B/C, HLA-DRB1*). Additionally, the authors identified cells showing high expression of genes associated with hypoxia (*HIF1A*, *VEGFA*). Surprisingly, only microglial cells were identified within the GAMs population, whereas monocytes and BM-macrophages were not distinguished. The sample size was relatively low (n = 1701 microglial cells) which could impede more detailed cell type identification and characterization of functional phenotypes. Using CyTOF, the authors confirmed the expression of HLA-DR, TREM2 (triggering receptor expressed on myeloid cells-2) and APOE (apolipoprotein E) in microglial cells (P2RY12+TMEM119+) [59]. Transmembrane protein 119 (TMEM119) is a reliable marker of microglia. However, expression of interferon-associated genes was not validated. Bulk transcriptomic analysis detected a type I IFN response in glioma BM-macrophages (CD49d+) but not in microglia (CD49d-) [78], pointing to the infiltrating GAMs as the major source of the interferon-related genes.

More detailed characterization of functional GAM phenotypes was provided in the recent scRNA-seq study on newly diagnosed (ND) and recurrent GBMs [70]. In line with previous reports, the authors identified three major functional transcriptomic profiles: (1) interferon-related associated with increased expression of *STAT1*, *IFIT2*, *ISG15*, *CXCL10*; (2) phagocytosis/lipid-related showing enhanced expression of *GPNMB*, *LGALS3*, *FABP5*, *CD9;* and (3) hypoxic characterized by induction of *BNIP3*, *ADAM8*, *MIF*, *HILPDA*. Those signatures were found in both ND and recurrent gliomas and were recapitulated in a murine glioma model (C57BL6 mice injected intracranially with GL261 cells). However, when microglia and macrophage GAMs were compared, macrophages more strongly activated the interferon and hypoxic signatures. Importantly, the identified transcriptomic signatures are in fact differentially enriched in different cell clusters that may encompass different cell populations. For the infiltrating myeloid cells, the authors defined monocytes, transitory monocyte-macrophages, and several macrophage clusters enriched in specific transcriptional programs. Similar monocyte-to-macrophage differentiation stages were identified in another work employing the same murine model [80]. Interestingly, in the mouse model the interferon-related genes *Rsad* (encodes type I protein secretion ATP-binding protein RsaD) and *Cxcl10* (encodes C-X-C motif chemokine ligand 10) also showed increased expression in the monocyte and transitory populations, whereas hypoxic and lipid/phagocytosis-related signature gene expression was augmented only in macrophages [70]. Interferon signaling is implicated into macrophage responses to pathogenic stimuli, known to elicit antiviral and immunoregulatory actions, and treatment with interferon has the anti-proliferative effect on tumor cells [reviewed in [81]. Genes of the lipid/phagocytosis-related signature were found to play tumor-supportive roles. GPNMB (glycoprotein Nmb), LGALS3 (galectin-3), and FABP5 (fatty acid binding protein 5) are implicated in inflammation, cell-matrix adhesion, and lipid metabolism, respectively. TREM2 expression was positively correlated with tumor progression, and the protein was implicated in promoting the immuno-suppressive TME [82]. TREM2 cooperates with CSF-1 in sustaining macrophage survival and proliferation [83]. ApoE is the best documented ligand of TREM2 [84] and CD9 is recognized as an anti-inflammatory marker of monocytes and macrophages [85].

Thus, interferon and lipid/phagocytosis signatures may yield opposite activities. CyTOF and scRNA-seq studies pointed to the possibility that anti-tumor monocytes migrate to tumors, where they differentiate to immunosuppressive macrophages and this transition is a consequence of tumor-induced education [69,80]. Possibly, monocyte-to-macrophage transition is connected with gradual changes of the transcriptional programs.

### 3.3. Immune Microenvironment of Gliomas Depends on the Tumor Genomic Background

We explored human glioma single-cell omics studies to find evidence for the assumption that distinct alterations in the genetic background dictate a specific immune TME and influence clinical outcomes. Malignant gliomas display considerable genetic and epigenetic heterogeneity, which affects patient survival and responses to therapy.

Combination of histology with molecular genotyping of key markers: IDH, ATP-dependent helicase (ATRX), Lys-27-Met mutations in histone 3 (H3K27M), TP53 mutations, and 1p/19q chromosomal deletion improved the classification of gliomas and provided some predictors of patient clinical outcome [4,86]. The transcriptional proneural, classical, and mesenchymal GBM subtypes show varied contribution of tumor-associated glial and microglial cells, and these subtypes associate with a specific TME [66]. Stratification of WHO grade II-IV glioma patients according to mutational burden indicates that a high number of mutations is associated with significantly worse overall survival and higher expression of immune checkpoint encoding genes [87,88]. Importantly, patients with high mutational burden showed the enrichment of immune cell signatures [89].

All these observations suggest a link between a genetic background of the tumor, immune cell composition, and potential responses to immunotherapies. In recent years several single-cell studies have helped to dissect the immune microenvironment of human gliomas and provided evidence for emerging associations between specific tumor somatic mutations and the composition of immune cells within glioma TME.

#### 3.3.1. Impact of IDH Status on the Immune Microenvironment of Gliomas

The *IDH* mutation status emerges as a potent modulator of the infiltration of immune cells in glioma TME. Friebel et al. (2020) used a mass cytometry analysis measuring 74 parameters to delineate the myeloid and lymphoid compartment of the brain tumor microenvironment [69]. They showed that GAMs constitute a major immune cell population encompassing up to 80% of all immune cells in the TME. In IDH-wt gliomas monocytes-derived macrophages (MDMs) comprise around 30% of GAMs, whereas in IDH-mut gliomas such cells occur at a very low number and microglial cells dominate the GAM population (Figure 1). These results were corroborated in a bulk RNA-seq study, in which microglia (CD49d^low^) and macrophages (CD49d^high^) were separated with FACS sorting [78]. FACS analysis also confirmed the exclusive expression of CD49d on CNS-invading myeloid cells.

Exhaustion, characterized in part by the upregulation of immune checkpoints, contributes to the T-cell dysfunction in GBMs. Levels of inhibitory receptors PD-1, LAG-3/CD223 (lymphocyte activation gene 3 product), TIGIT (T cell immunoreceptor with immunoglobulin and ITIM domain), and ectonucleotidase CD39, combined with T-cell hyporesponsiveness of tumor-specific T cells, suggest poor function of tumor infiltrating lymphocytes (TILs) and confirm severe exhaustion observed in GBM [90]. These observations have been corroborated by the analyses of CD45+ immunosorted cells from human gliomas. Significant increases in myeloid cells in *ID*H-mut and *ID*H-*wt* gliomas, and lymphocytes in *ID*H-*wt* tumors, have been reported. Multicolor FACS analyses of 14 major immune cell populations across 100 clinical samples followed by RNA-seq of samples from 48 patients demonstrated the differential abundance of microglia and BM-derived macrophages in IDH-mut and IDH-wt gliomas [78]. Spatial analysis of tissue sections showed the enrichment of GAMs in the perivascular niche and a closer proximity of MDMs to the vessels compared with activated microglia. Consistently, examination of the transcriptomic data from the Ivy Glioblastoma Atlas Project demonstrated the enrichment of MDMs in the microvascular compartment, in which CD4+ and CD8+ T cells were found in *IDH* WT gliomas [78].

Application of CyTOF (besides dissection of microglia and BM-derived macrophages) revealed changes in other immune cell populations. Among tumor-infiltrating immune cells, Friebel et al. (2020) identified T cells (CD3^+^), B cells (CD19^+^HLA-DR^+^), NK cells (CD56^+^CD16^+^), neutrophils (CD66b^+^), two subsets of classical DCs: cDC1 (CD141^+^CADM1^+^) and cDC2 (CD1c^+^) plasmacytoid DCs: pDCs (CD123^+^), and plasma cells (CD19^+^, CD38^high^). There was no difference in the overall T cell infiltration rate between IDH variants; however, Tregs were significantly more frequent in IDH-wt gliomas and T-cell frequency positively correlated with pDCs and cDCs frequencies, whereas increased numbers of all those populations were associated with decreased GAM/monocyte frequencies [69]. The *MGMT* promoter methylation status did not change frequencies of major immune populations [69].

In addition to the reduced number of TILs and macrophages, IDH-mut gliomas show lower PD-L1 expression compared to IDH-wt gliomas [91,92]. A proportion of lymphocytes (T-cells, B-cells, NK cells) was low in IDH-mut gliomas (~10%), whereas in IDH-wt lymphocytes constituted 25% of all immune infiltrating cells [78]. Additionally, division of T cells into five functional subsets: naïve, central memory (CM), effector memory (EM), terminally differentiated effector memory (TEMRA), and non-circulating tissue-resident (RM), according to the expression of five discriminatory markers (CD45RA, CD45RO, CCR7, CD127, and CD103) [93], allowed assessing more subtle differences in the T cell populations. Friebel et al. (2020) reported that the majority of T cells found in GBMs were memory T cells [69]. CD8 RM and EM T-cells had lower expression of proliferation and activation markers in IDH1^mut^ compared to IDH1^wt^ gliomas. In contrast, the IDH status had no effect on expression of co-stimulatory (ICOS, CD27, and CD137) and co-inhibitory receptors (2B4, TIGIT, and PD-1) [69].

The oncometabolite 2-HG inhibits complement activation, complement-mediated phagocytosis, and migration of activated T-cell, their proliferation, and cytokine secretion [94]. Reduced PD-L1 expression might be a result of lower infiltration of PD-L1 expressing immune cells, but could emerge from epigenetic silencing of the immune checkpoint genes in glioma cells, due to 2-HG-driven DNA hypermethylation [95,96]. Studies of mouse syngeneic glioma models indicated the lower expression of cytokines in IDH-mut gliomas that might reduce leukocyte chemotaxis [92]. Still, the mechanism underlying lower infiltration of GAMs in IDH-mut gliomas remains to be elucidated.

Innate lymphoid cells play in anti-tumor immunity and are regulators of TME [97]. The analysis of NK cells (CD56^+^CD3^-^) showed that the two main populations of CD56^int/bright^CD16^−^ and CD56^int^CD16^+^ correspond to the immature and the cytotoxic NK cells, respectively. The enrichment of immature CD56^int/bright^ was found in CD16^−^ NK cells among lymphocytes in the IDH1-wt gliomas, whereas predominantly CD56^int/bright^CD16^+^ NK cells accumulated in the IDH1-mut gliomas. Splitting of the IDH1-wt glioma cohort according to the *MGMT* promoter methylation showed a trend toward a higher proportion of CD56^int/bright^CD16^+^ NK cells in the unmethylated cases. Frequencies of immature NK cells negatively correlated with overall survival in the IDH1-wt cohort (but not significantly) [69].

Mathewson et al. (2021) investigated T-cell subtypes and expression programs across IDH variants, specifically in glioblastomas [72]. ScRNA-seq profiling of T cells from 26 IDH-wt and IDH-mut GBMs revealed the presence of CD8 T cells, CD4 conventional T cells (CD4 Tconv), CD4 Tregs, and cycling T cells. Interestingly, the corticosteroid therapy with dexamethasone was associated with substantially reduced numbers of infiltrating T cells (mean reduction was 4.14- and 7.72-fold for CD3+ and CD4+ T cells, respectively). The overall representation of clusters was similar in IDH-wt and IDH-mut GBMs. CD8 and CD4 T cells expressed an interferon signature, an effector memory signature, or a stress signature. The latter was not an artefact, as expression of these genes was detected by RNA in situ hybridization for glioma-infiltrating CD3E+ T cells [72]. T cell-specific genes (including cytotoxicity genes *PRF1* (*Perforin 1*) and *GZMA* (*Granzyme a*) were more highly expressed in IDH-wt GBMs than IDH-mut GBMs. The cytotoxicity score of CD8 T cells (*PRF1, GZMB, GZMA, GZMH, NKG7, GNLY*) was associated with an increased signature of NK cells (*KLRD1*, *FGFBP2*, *FCGR3A*, *S1PR5*, *KLRC1*, *KLRC3*, *KLRB1*, *KLRC2*). Several NK cell receptor genes, including *KLRC2* (NKG2C protein), *KLRC3* (NKG2E protein), *KLRC1* (NKG2A protein), *KLRD1* (CD94 protein), and *KLRB1* (CD161 protein) were expressed by CD8 T cells with high cytotoxicity scores. These cells represent effectors that share transcriptional profiles with innate T cells, despite having a diverse TCR repertoire. High cytotoxicity of CD8 T cells correlated with lower expression of the *PDCD1* gene (PD-1 protein) and genes coding for co-inhibitory receptors (CTLA4, HAVCR2, LAG3, and TIGIT). The highest level of *PDCD1* was found in cytotoxic CD4 T cells, whereas *TIGIT* was most highly expressed by Tregs. The genes coding for the inhibitory CD161 receptor (*KLRB1*) and the activating NKG2C/CD94 receptor (*KLRC2* and *KLRD1*) were expressed by a large fraction of CD8 T cells. *KLRB1* was preferentially expressed by CD8 and CD4 Tconv cells in diffuse gliomas; its expression was lower in CD4 Tregs. The authors postulated that expression of NK cell receptors is induced in T cells by inflammatory mediators in the glioma TME. CLEC2D, the ligand for CD161, is a surface molecule expressed by immunosuppressive myeloid cells and malignant cells. Mechanistic in vitro and in vivo studies demonstrated that the CD161 receptor inhibits T cell function, including cytotoxicity and cytokine secretion [72].

#### 3.3.2. The Effects of Co-Deletion of 1p/19q in IDH Mutant Gliomas

Venteicher et al. (2017) performed scRNA-seq analysis of IDH-mut astrocytomas (IDH-A) characterized by *TP53* and *ATRX* mutations, and oligodendrogliomas (IDH-O) characterized by mutations in the *TERT* (*Telomerase reverse transcriptase*) gene promoter and co-deletion of chromosome arms 1p and 19q (1p/19q) [13]. The authors confirmed the genetic background of the investigated samples at single-cell level. Most genes with higher expression in cells from IDH-A were encoded in 1p/19q regions that are deleted in IDH-O, whereas TP53 targets were enriched in IDH-O as compared with *TP53* mutated IDH-A. The 1p/19q region contains genes encoding potent immunoregulatory proteins including *CSF1* encoded in a 1p region and *TGFB1* encoded in the 19q region. CSF1 is involved in proliferation, differentiation, and survival of myeloid cells and TGFβ1 is a potent immunosuppressive cytokine. Thus, a lack of these genes might negatively influence accumulation of microglia and macrophages in glioma TME. Consistently with this notion, the estimated relative abundance of microglia/macrophages was higher in IDH-A tumors compared to IDH-O tumors [13].

A recent CyTOF study of the composition of immune infiltrates in IDH-A and IDH-O tumors showed that IDH-A have increased levels of VEGF and TGFβ that play tumor-supportive roles. However, no difference in the proportion of glioma-associated macrophages was noted [67]. These observations have been supported by the results of CIBERSORT deconvolution of the TCGA dataset combined with IHC staining of LGG samples, that demonstrated a lower number of “M2-related” markers in gliomas with 1p/19q co-deletion [73]. Co-deletion 1p/19q is a strong, good prognostic marker. Better survival of patients harboring this alteration might be related with reduced infiltration of GAMs, as deletion of the 1p region that encodes the *CSF1* gene yields similar survival outcome as full 1p/19q co-deletion, whereas the effect of 19q deletion is marginal [98]. The identified transcriptional programs of GAMs, their functional states, and localization in GBMs are depicted in the Figure 2.

#### 3.3.3. Immune Microenvironment in the Molecular Subtypes of GBM

A number of studies demonstrated that infiltration of GAMs increases with glioma grade and is a negative prognostic marker [13,49,99]. IDH-wt GBMs have been classified into molecular subtypes based on their genetic and transcriptional profiles [10], and the influence of immune infiltrates on the GBM-intrinsic gene expression has been noted [66]. The mesenchymal subtype (MES) shows the worst overall survival and was found to be associated with *NF1* mutations/deletions and high abundance of macrophages [66]. The NF1 deficiency promotes macrophage infiltration, which was evidenced by increased accumulation of AIF1+ (Allograft inflammatory factor 1) cells in the proximity of NF1-cells, as compared with NF1+ cells in human GBMs and the decreased rate of cell infiltration in *NF1* knock-down cell cultures [66]. Additionally, the deletion of 5q chromosome arm was negatively correlated with the MES-like cell state. Genes encompassed in this region encode for a number of cytokines—*CSF2*, *IL2*, *IL4*, *IL5*, *IL13* and *CXCL14*—that could be involved in the communication between microglia/macrophages and the MES-like tumor cells [12]. The CIBERSORT analysis indicated that mesenchymal GBMs show the enriched gene signature of “M1” macrophages, “M2” macrophages, and neutrophils, whereas the activated NK signature is reduced. The proneural subtype GBMs, which show the best prognosis among the three molecular subtypes, had significantly lower signature of CD4+ T-cells and the classical GBM showed significantly stronger dendritic cell signature [66].

Hara et al. (2021) performed an extensive investigation of the cross-talk between the immune microenvironment and the mesenchymal GBM subtype [71]. Using a mouse glioma model induced via transduction with the lentivirus harboring GFP, Hras^G12V^ and sh-p53, the authors confirmed a presence of the tumor cell states consistent with the ones observed in human glioblastomas [12]. A fraction of the MES-like cells, defined as GFP+, PDPN (Podoplanin)+, PDGFRα- cells, was proportional to the abundance of the CD45+ immune cells. Immunohistochemical analysis showed an enrichment of the IBA1+ macrophages in the vicinity of the MES-like cells in the mouse model. The finding was corroborated in a human glioblastoma sample, in which cell types were identified with a panel of 135 genes measured by robust multiplexed fluorescence in situ hybridization (MERFISH). Moreover, the association of the MES-like state and macrophages appeared to be causal, as macrophage depletion with clodronate prior to tumor transplantation led to significant reduction of the MES-like cells in the tumor. In search of the molecular mechanism driving this association, the authors identified receptor-ligand pairs that could be involved in communication between MES-like tumor and macrophage cells. They demonstrated that macrophages induce the MES-like glioblastoma cell state via the macrophage-derived oncostatin M (OSN) and its receptors (OSNR and LIFR) expressed by glioblastoma cells [71]. This regulatory loop is a potential therapeutic target in MES-GBMs.

#### 3.3.4. The Immune Microenvironment of Recurrent Gliomas

Glioblastomas are highly lethal tumors because regardless of intensive treatment (including surgery, radiotherapy, and chemotherapy), in the majority of cases a primary tumor relapses in a therapy-resistant form. It was reported that upon recurrence tumor cells are able to switch their molecular profiles into a more aggressive one. Glioblastomas may switch their molecular subtype upon recurrence and most frequently they transit to the MES and PN subtypes [66]. Interestingly, a bulk RNA-seq study combined with CIBERSORT deconvolution on 91 matched pairs of primary and recurrent GBM demonstrated that the immune cell fraction is associated with non-MES/MES transition. Tumors switching from non-MES to MES and MES to non-MES at recurrence showed an increase and decrease of the immune cells, respectively. Accordingly, a predicted frequency of “M2” macrophages was significantly higher at recurrence in cases transitioning to the MES subtype [66]. These results were corroborated in a scRNA-seq study showing that the difference between macrophage scores in initial and recurrent tumors correlated positively with the MES-like score, but was inversely correlated with the scores for other expression programs (OPC-, NPC-, AC-like).

Interrogation of composition of cells in TME after treatment and at tumor recurrence using scRNA-seq showed substantial differences [70] and these changes might contribute to overall acquired resistance to therapy. While in newly diagnosed gliomas, GAMs are the most abundant immune cells in TME, after recurrence the immune compartment of TME becomes more diverse, with a higher percentage of lymphoid cells, including T cells, NK cells, and B cells [70]. Analysis of the ontogeny of GAMs present in the microenvironment showed that after recurrence the majority of GAMs in TME were monocyte-derived cells (especially under hypoxic conditions), in contrast to newly diagnosed tumors where microglia-derived GAMs were more abundant. Within the group of primary tumors, microglia-derived GAMs constituted the major population (34–75%), whereas macrophage-derived GAMs were less abundant (16–45%) and T-cell infiltration was relatively low (2–20%). The immune cell composition was found to change significantly upon recurrence, as T-cells were the most abundant group (33–75%), and the macrophage-derived GAMs (5–39%) outnumbered microglia-derived GAMs (1–17%), which, in contrast to primary tumors, constituted a rather minor population in the recurrent tumors. Differential gene expression analysis of selected GAM subpopulations in newly diagnosed versus recurrent tumors showed that in cells from recurrent samples genes related to monocyte chemotaxis, IFN signaling and phagocytosis were significantly upregulated. Interestingly, the authors identified common gene expression signatures of subsets of GAMs present both in newly diagnosed and recurrent tumors, as well as in murine experimental gliomas. These signatures consisted of genes associated with phagocytosis and lipid metabolism, hypoxia, and genes known to be interferon-induced [70].

Similarly, a CyTOF analysis of 13 primary and 3 recurrent GBMs showed that the proportion of GAMs among CD45+ tumor-infiltrating cells decreases upon recurrence [68]. Nevertheless, simultaneous measurement of 30 parameters was not sufficient to reliably discriminate microglia and macrophages, which prevents estimation of the brain-resident and peripheral GAMs. Moreover, the authors were not able to identify a large proportion of cells from recurrent tumors (58%) [68]. Providing proper cell type identification could affect the obtained results. Acquiring g more information about the immune cell composition of the recurrent gliomas from the longitudinal studies using single-cellomics would greatly improve our understanding of the evolution of the glioma TME.

## 4. Challenges and Perspectives

While single-cell omics allowed for the unprecedented increase in understanding heterogeneity and diversity of cell populations within glioma TME, there are some challenges of both technologies that need to be further addressed:CyTOF allows for protein measurement and identified populations better reflect the immunophenotyping capacities that are usually limited to a smaller set, mostly surface markers. Still, caution should be taken in the interpretation of the CyTOF studies, as the ability to discriminate discrete populations is largely affected by the supervised (expert) selection of a limited number of parameters that are measured. Currently, sets of limited markers do not allow us to fully characterize functional states of cells and to detect underlying molecular mechanisms.Despite the fast development of CyTOF-dedicated analysis methods, there is still no “gold” standard of the preceding standardization of analytical procedures (data preprocessing). Various sources of technical variations in CyTOF have been identified, such as differences in the instrument sensitivity, change in oxidation rate during long-term sample running that may cause signal fluctuations, and the interference artifacts between mass detection channels [100]. Moreover, some analysis methods were adapted from flow cytometry data analysis workflows where the plots are typically used for gating (the manual assignment of cells to cell groups) with data randomization for visualization of bivariate distributions. In CyTOF the randomization settings are not reported, making the re-analysis of data difficult [101]. It is recommended that CyTOF studies should provide their raw data and a precise description of all preprocessing steps to ensure replicability, re-usability, and the correctness of future analysis [100].scRNA-seq appears to be more reproducible across laboratories. As data deposition in public repositories becomes widespread, re-analyses of the datasets can help to compare data from different studies or validate findings from animal models in human samples. Some studies provide access to the interactive datasets through web applications, which can be used without advanced programming skills (https://singlecell.broadinstitute.org/single_cell, https://www.brainimmuneatlas.org/). Still, precaution should be taken when cell populations are identified based on scRNA-seq data. Cell clusters frequently used to describe scRNA-seq results do not necessarily correspond to cell population and number of cell clusters can be regulated by adjusting clustering parameters. The observed clusters may as well represent different states of the same cell type.scRNA-seq allows for the identification of a vast number of differentially expressed genes and recognition of cell/state specific signaling pathways and gene networks. However, RNA expression may not correspond to the protein level [63]. Thus, expression of individual genes demarcating specific populations should be validated on a protein level and identification of functional state should be confirmed by a comprehensive biochemical signature overlapping multiple parameters.Both CyTOF and scRNA-seq mainly rely on cell isolation from the original setting during which cells are isolated from their local niches and these “snapshots” lose spatial information regarding cell position and interacting cells. Current technologies allow us to acquire positional information by integrating imaging and positional barcoding information. Spatial transcriptomics provides information about tissue architecture-dependent as well as position-dependent cellular functions. Recently introduced *10X Genomics* Visium (https://www.10xgenomics.com/products/spatial-gene-expression/), which employs spatial transcriptomics using barcode-based approaches, and CARTANA (http://cartana.se), based on in situ sequencing, allow us to capture tissue-specific, spatial organization of gene expression.

Localizing cells of a particular type or with a transcriptional program may help to answer pressing questions regarding specific roles of various immune cells found in the TME. Nevertheless, an increasing number of single-cell omics techniques comes with the need to employ advanced bioinformatic analyses that require specialized professionals.

## 5. Conclusions

Single-cell omics studies verified many previous assumptions and provided strong evidence for the anticipated diversity of myeloid and lymphoid infiltrates in the glioma TME. A ratio of tumor infiltrating microglia and monocytes-derived macrophages is different in IDH-wt and IDH-mut gliomas, with the predominance of microglia and lower infiltration of lymphocytes in the later. This suggests that cytokines/chemokines released by immunomodulatory macrophages may drive T-cell enrichment in TME of IDH-wt gliomas. While the abundance of CD4+ and CD8+ T-cells did not show major differences between the IDH variants, regulatory T-cells were more frequent in IDH-wt [69]. Infiltration of T-cells, B-cells, and NK cells was low in IDH-mut gliomas, likely due to the inhibitory effect of 2-HG. An increased number of Tregs found in IDH-wt gliomas might result in the suppression of cytotoxic CD8 T cell responses. Interestingly, there was an enrichment of immature CD56^int/bright^CD16^−^ NK cells among lymphocytes in the IDH1^wt^ gliomas. A highly immunosuppressive GBM microenvironment can alter the NK cell phenotype and functions resulting in their inability to effectively carry out anti-tumor activities.

Across molecular GBM subtypes, the MES-like state is associated with increased overall T-cell accumulation. This observation is supported by the fact that the MES-like score calculated for the TCGA dataset showed stronger correlation with cytotoxicity markers (*GZMB*, *PRF1*), T-cell exhaustion markers (*FOXP3*, *LAG3*, *PDCD1*, *TIGIT*), and overexpression of the antigen presentation complexes MHC-I and MHC-II in MES-like GBMs compared to other molecular subtypes [71]. T-cell enrichment can result in both pro- and anti-tumor activity, depending on the functional state of T-cells. Regulatory T-cells can be an indicator of immunosuppression, whereas the presence of cytotoxic CD8+ T-cells can point to more effective anti-tumor activity. However, a number of Tregs in tumor tissues failed to predict patient’s outcome, showing moderate or no correlation with survival [102]. Further studies of discrete subpopulations are required to define their impact in anti-tumor immunity. The existence of distinct Treg subsets which may differ in their suppressive capacity or their different ratio in tumors may vary between individual patients. Future therapeutic interventions may target specific subpopulations with recognized immunosuppressive roles, thus assessing composition and functionality of the immune cell types in glioblastoma with a specific genetic background, are of high importance.

A better knowledge of functional characteristics and spatial distribution of immune cells within the immune compartment of gliomas would provide insights into the complex cellular and molecular networks that determine the immunosuppressive states in glioblastomas. This, as a result, may also generate the next molecular targets for therapeutic intervention.

## Figures and Tables

**Figure 1 cells-10-02264-f001:**
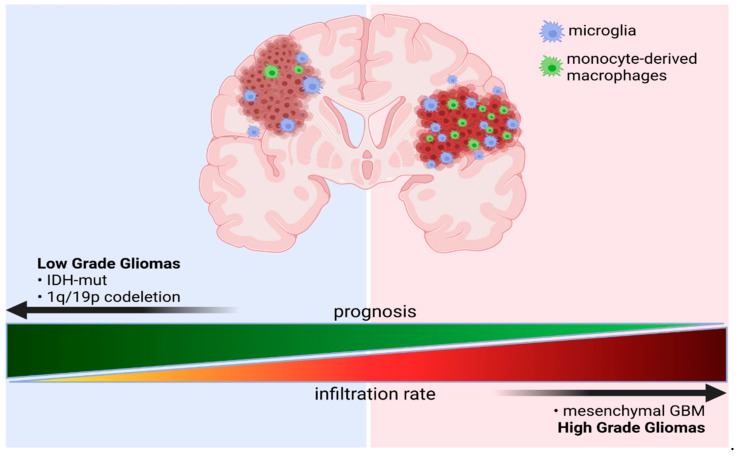
Schematic representation of a ratio of microglia and monocyte-derived macrophages (MDMs) in gliomas of different grades and with distinct genetic backgrounds. *Created with BioRender*.

**Figure 2 cells-10-02264-f002:**
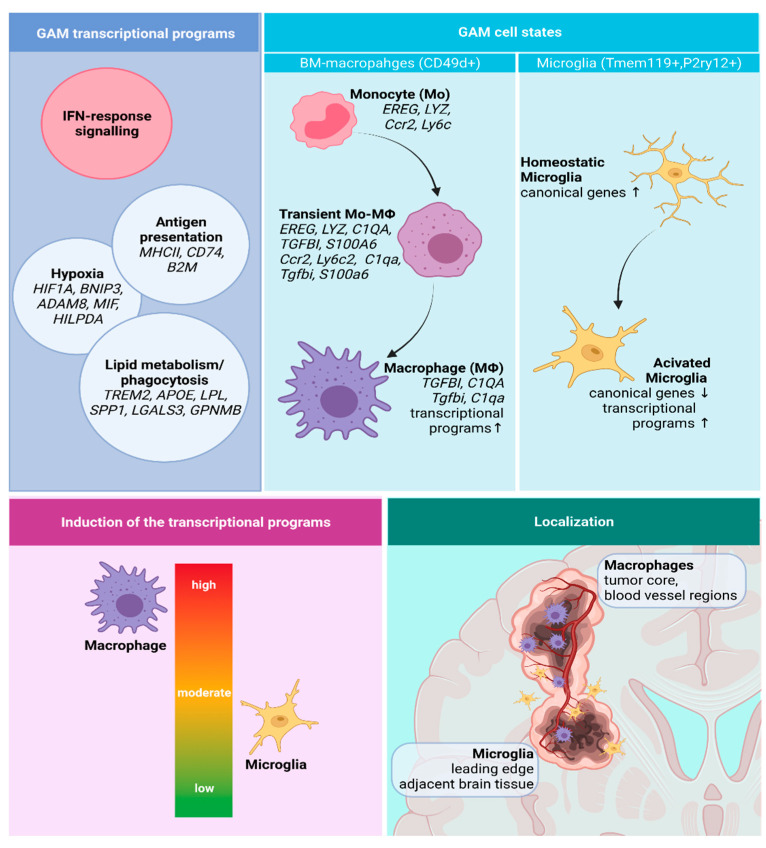
A summary of the identified transcriptional programs, cell states, and spatial distribution of GAMs. Created with Biorender.

**Table 1 cells-10-02264-t001:** List of the reviewed scRNA-seq and CyTOF publications on human malignant gliomas.

References	Glioma Type	Species	Methodology
Darmanis et al. 2017 [56]	IDH-wt GBMs	Human	scRNA-seq
Wang et al. 2017 [66]	IDH-wt GBMs, molecular GBM subtypes, paired primary and recurrent	Human	scRNA-seq (on tumor cells), CIBERSORT
Venteicher et al. 2017 [13]	WHO grade II-IV gliomas, IDH-mut	Human	scRNA-seq
Müller et al. 2017 [57]	WHO grade II-IV glioma	Human and mouse	scRNA-seq
Neftel et al. 2019 [12]	IDH-wt GBMs, pediatric and adult, primary and recurrent	Human and mouse	scRNA-seq
Sankowski et al. 2019 [59]	IDH-wt GBMs	Human	scRNA-seq, CyTOFF
Fu et al. 2020a [67]	WHO grade II gliomas, IDH variants, diffuse astrocytoma and oligodendroglioma	Human	CyTOFF
Fu et al. 2020b [68]	GBMs, IDH variants, primary and recurrent	Human	CyTOFF
Friebel et al. 2020 [69]	WHO grade II-IV gliomas, brain metastases, IDH-variants	Human and mouse	CyTOFF
Antunes et al. 2021 [70]	IDH-wt GBMs, primary and recurrent	Human and mouse	scRNA-seq, CITE-seq
Hara et al. 2021 [71]	IDH-wt GBMs, primary and recurrent	Human and mouse	scRNA-seq
Mathewson et al. 2021 [72]	GBMs, IDH variants	Human and mouse	scRNA-seq
Zhang et al. 2021 [73]	WHO grade II/III glioma, 1p/19q co-deletion variants	Human	scRNA-seq

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
