# Peer review of "Single-Cell Omics in Dissecting Immune Microenvironment of Malignant Gliomas—Challenges and Perspectives"

_cells, 2021, doi:10.3390/cells10092264_

Round 1

Reviewer 1 Report

The manuscript by Dr. Kaminska and colleagues aims to review the advances that single cell-omics have brought in our understanding of the immune microenvironment of malignant gliomas. This is a very ambitious manuscript, which reviews the topic extensively and also includes background on malignant glioma biology, CNS immune environment in development and neuropathology, glioma immune microenvironment, as well as single cell omics studies related to the above. Recent milestone reports regarding the main topic are covered and the authors have included passages, tables and illustrations with interesting original contribution.

However, the endeavor to discuss all the topics listed above makes the manuscript difficult to read. The text reads like a list of facts and the most salient points are hard to identify. Maybe the authors could consider some of the following points:

1) What is the audience for this manuscript? Readers interested in what single-cell omics have taught us about the immune microenvironment in malignant gliomas very likely know the basic facts about gliomas and their molecular classification. Part 1.1 could be significantly shortened.

2) The target readers also likely know the basics of the immunological uniqueness of the central nervous system. Part 1.2 includes a mixture of decade-old studies with some interesting and recent single cell studies. Unfortunately, the single-cell studies are only cited as a generic statement that provides little concrete information (e.g. ,“Recent fate-mapping and multi-omics studies have consistently demonstrated that microglia and BAMs are distinct cellular entities in respect of their transcriptomic and protein expression profiles, ability to self-renew, and their development [21–25]” , page 4). Maybe this part could be shortened or structured to highlight the results of single cell analyses.

3) Many paragraphs are half-page long (e.g. p8, p11). Adding additional subtitles and breaking the paragraphs down into smaller pieces would improve readability.

4) As is, original contributions such as “We explored human glioma single-cell -omics studies to find evidence for the assumption that distinct alterations in the genetic background dictate a specific immune TME and influence clinical outcomes.” are lost within the text. Such statements of intent would be very helpful at the beginning of each part, with the subsequent text organized to explain or illustrate the authors’ point of view. Also, please accentuate which facts/ references you consider especially important.

5) Please number the lines.

Author Response

  1. What is the audience for this manuscript? Readers interested in what single-cell omics have taught us about the immune microenvironment in malignant gliomas very likely know the basic facts about gliomas and their molecular classification. Part 1.1 could be significantly shortened.

Ad. We agree that the manuscript is primarily addressed to neuro-oncologists who are familiar with the facts regarding classification or genetic alterations of gliomas. In general, we followed the reviewer 1 comments to make a description of gliomas shorter. We reduced the content of this chapter leaving only necessary information in the revised text However, we believe that this review would be of interest to all researchers working on TME, including those studying non-CNS tumors, therefore we insist that a brief summary or reminder of genetic alterations in gliomas, their classification and peculiarities due to specific features of the CNS environment, is important. As we have tried to correlate immune TME with genetic background of tumors, we found it necessary to describe it briefly (pelase consult the tracked changes version).

2) The target readers also likely know the basics of the immunological uniqueness of the central nervous system. Part 1.2 includes a mixture of decade-old studies with some interesting and recent single cell studies. Unfortunately, the single-cell studies are only cited as a generic statement that provides little concrete information (e.g. ,“Recent fate-mapping and multi-omics studies have consistently demonstrated that microglia and BAMs are distinct cellular entities in respect of their transcriptomic and protein expression profiles, ability to self-renew, and their development [21–25]” , page 4). Maybe this part could be shortened or structured to highlight the results of single cell analyses.

Ad. We followed the reviewer 1 comments to make a description of CNS environment shorter by removing older concepts. We have published recently in Inter. Journal of Molecular Sciences a review describing how sc-omics studies expanded our understanding the functions of brain macrophages under steady state conditions and there are other excellent reviews referring to this aspects based on lineage tracing studies. As we focused on gliomas, we only briefly referred to this aspect and relevant references, to avoid repeating ourselves.

3) Many paragraphs are half-page long (e.g. p8, p11). Adding additional subtitles and breaking the paragraphs down into smaller pieces would improve readability.

Ad.  We thank for this comment on the text organization. We have added an additional subtitles 3.2.1. GAMs origin and localization influence the expressed phenotype; 3.2.2. Transcriptional programs of glioma-associated macrophages and divided the text into shorter paragraphs.

4) As is, original contributions such as “We explored human glioma single-cell -omics studies to find evidence for the assumption that distinct alterations in the genetic background dictate a specific immune TME and influence clinical outcomes.” are lost within the text. Such statements of intent would be very helpful at the beginning of each part, with the subsequent text organized to explain or illustrate the authors’ point of view. Also, please accentuate which facts/ references you consider especially important.

Ad. We modified the text to make it more clear and concrete. In our opinion we accentuated which facts/ references we consider especially important, as we discussed all published sc-omics studies and prepared the essence or verified part of them. As they are hardly comparable from technical point of view, a number of analyzed cells, discretization of phenotypes, various tumors primary/secondary, some compared to metastatic brain tumors, it is impossible to prioritize them. We found all them informative and what we present is the interpretation derived from all of them. That specific statement (We explored human glioma single-cell -omics studies to find evidence for the assumption that distinct alterations in the genetic background dictate a specific immune TME and influence clinical outcomes) has been moved at the beginning of the chapter.

5) Please number the lines.

Done.

Reviewer 2 Report

The manuscript is well written and well organised, with high-quality figures. Congratulations to the authors on a high-quality document.

I think the introduction could be shortened slightly (i.e. the introduction of glioma subtypes) 

I have attached a PDF doc with a few words/phrases highlighted that need correction 

Author Response

Thank you for appreciation of our review and indicating the mistakes/errors. We addressed in details all referee’s comment and made amendments in the revised text.

Reviewer 3 Report

The review “Single-cell omics in dissecting immune microenvironment of malignant gliomas – challenges and perspectives” is professionally written. Illustrations are  very informative and attractive (the authors very successfully used BioRender). The manuscript is covering in details single-cell technologies that allow analysis of glioma at the single-cell level. In the review, the authors summarize the findings based on a single cell analysis (RNA sequencing  and mass cytometry) and the current state of knowledge on a functional diversity of immune infiltrates in malignant gliomas with different genetic backgrounds. The problems with data obtained by this analysis were also discussed.

    The manuscript can be published in the journal “Cells”.

Author Response

Thank you for appreciating our review.

Round 2

Reviewer 1 Report

The authors have addressed all the reviewer's concerns. The manuscript is a comprehensive review of the topic and will be of interest to readers in several fields, especially neuro-oncology.